# Type and Frequency in Use of Nutraceutical and Micronutrient Supplementation for the Management of Polycystic Ovary Syndrome: A Systematic Scoping Review

**DOI:** 10.3390/biomedicines11123349

**Published:** 2023-12-18

**Authors:** Nicole Scannell, Evangeline Mantzioris, Vibhuti Rao, Chhiti Pandey, Carolyn Ee, Aya Mousa, Lisa Moran, Anthony Villani

**Affiliations:** 1School of Health, University of the Sunshine Coast, Sunshine Coast, QLD 4556, Australia; nicole.scannell@research.usc.edu.au; 2UniSA Clinical & Health Sciences, Alliance for Research in Exercise, Nutrition, and Activity (ARENA), University of South Australia, Adelaide, SA 5000, Australia; evangeline.mantzioris@unisa.edu.au; 3NICM Health Research Institute, Western Sydney University, Penrith, NSW 2145, Australia; v.rao@westernsydney.edu.au (V.R.); c.ee@westernsydney.edu.au (C.E.); 4Monash Centre for Health Research and Implementation (MCHRI), School of Public Health and Preventive Medicine, Monash University, Melbourne, VIC 3800, Australia; chhiti.pandey@monash.edu (C.P.); aya.mousa@monash.edu (A.M.); lisa.moran@monash.edu (L.M.)

**Keywords:** nutraceuticals, micronutrients, polycystic ovary syndrome, pcos, supplements

## Abstract

Lifestyle strategies are considered first-line treatment for the management of polycystic ovary syndrome (PCOS). However, complementary therapies, including nutrient supplementation, have been identified as a potential adjunct therapy. Therefore, we systematically mapped the available literature to identify the type and frequency of the use of nutraceutical and micronutrient supplementation for the management of PCOS features. A systematic search of the literature was conducted using CINAHL, Cochrane reviews, Medline, PsycINFO, Scopus and LILACS. All types of study designs were included if they reported on the use of nutraceuticals and/or micronutrient supplementation on features of PCOS in women aged ≥18 years. A total of 344 articles were included. Forty-one supplements were identified, with the most frequently investigated being inositols (n = 86), vitamin D (n = 53), N-acetylcysteine (n = 27) and omega-3 fatty acids (n = 25). Reproductive outcomes were the most commonly reported (n = 285; 83%), followed by metabolic (n = 229; 67%), anthropometric (n = 197; 57%) and psychological (n = 8; 2%). Our results identified that nutraceutical and micronutrient supplementation require further investigation of psychological outcomes in women with PCOS. Moreover, adequately powered primary studies are warranted to investigate therapeutic doses needed for clinical benefits.

## 1. Introduction

Polycystic ovary syndrome (PCOS) is one of the most common hormonal conditions experienced by women of reproductive age, affecting around 12% depending on the population studied and diagnostic criteria applied [1]. Women with PCOS experience a broad range of clinical symptoms that can be collectively grouped into anthropometric, metabolic, reproductive and psychological manifestations, which can change and develop throughout the lifespan [2,3]. Typically, clinical features used for the diagnosis of PCOS in adults include abnormal menstrual cycles, hypoandrogenism and polycystic ovaries. Most recently, the 2023 International PCOS guidelines recommend the presence of at least two of the three following features: oligo-/anovulation, the identification of hyperandrogenism using either clinical examination or biochemical testing and polycystic ovarian morphology identified via ultrasound or testing of anti-Mullerian hormone [1]. However, the health implications for women with PCOS are much broader, stemming from key pathophysiological features, including insulin resistance and hyperandrogenism [2,3,4]. Women with PCOS experience intrinsic insulin resistance, which is mechanistically distinct from that associated with excess weight [4,5,6,7]. Moreover, these women have a greater propensity for weight gain, which further worsens features of PCOS, including insulin resistance [8,9]. Accordingly, women with PCOS have a greater prevalence of central adiposity [8,9] and are at greater risk of cardiometabolic diseases, including cardiovascular disease (CVD), metabolic syndrome and type 2 diabetes [3,10]. Moreover, PCOS is also a leading cause of anovulatory infertility [11] and is associated with adverse pregnancy and neonatal outcomes, including miscarriage, pre-eclampsia, premature delivery and gestational diabetes compared with healthy matched controls [1,12]. Lastly, women with PCOS are also more vulnerable to mood disturbances, including anxiety, stress and depression [13,14,15], as well as disordered eating symptomology [14,16].

Given the chronic nature of PCOS, the importance of identifying effective strategies to ameliorate and manage its symptoms is of clinical importance for both short and long-term health. At present, lifestyle interventions, including adopting a healthy diet and regular physical activity, are considered first-line treatment for the management of PCOS, in particular for weight management and ameliorating CVD risk [17]. Nevertheless, as presented in the 2023 International PCOS Guidelines [17], there is limited high-quality evidence to support one particular dietary approach as part of lifestyle management for women with PCOS beyond that of a healthy dietary pattern consistent with general population-based dietary guidelines. Although lifestyle changes are considered the first-line treatment for PCOS management, pharmacological agents are also often used [18]. Pharmacological therapies can be an adjunct to lifestyle modification interventions and can be prescribed to improve reproductive and metabolic features [19]. However, as individuals with PCOS often experience an expansive range of symptoms that can differ for each person, identifying standardised interventions to manage its complexities remains challenging [20]. Therefore, alternative or additional management strategies may still be sought by some women [21,22].

Supplementation with nutraceuticals or specific micronutrients has been identified as an adjunct therapeutic approach to lifestyle and/or pharmacological agents for the treatment of chronic diseases, including PCOS [23]. Despite no universally accepted definition for the term ‘nutraceuticals’, they have previously been described as nutrient derivatives or extracts of plants, animals or foods that may elicit a pharmacological response [24]. Furthermore, there is evidence suggesting that women with PCOS intentionally seek out complementary medicines (CM) for the management of PCOS symptoms [21,22]. A previous cross-sectional analysis of 493 Australian women with PCOS reported that more than 70% of the sample had used some form of CM in the previous 12 months, with micronutrient supplementation including vitamins, minerals, fish oil capsules and herbal supplements being the most commonly reported [21]. Further, almost one-third of these women perceived the therapeutic benefits of CMs for managing PCOS symptoms, with few women reporting adverse reactions to nutritional and herbal supplements [21].

Previous systematic reviews on the efficacy of nutraceutical and micronutrient supplementation in ameliorating PCOS symptoms have reported inconsistent findings [25,26]. Moreover, heterogeneity in study designs, sample sizes, participant cohorts and adequate control methods have also been identified as potential limitations [27]. To date, clinical practice guidelines and synthesis of higher levels of evidence have been reported on the efficacy of multimodal lifestyle interventions (diet and physical activity) for the management of PCOS, with no specific reference or recommendations for specific nutrients [18,28]. Given that the use of nutritional supplements for the management of PCOS features is rapidly growing, synthesis of higher levels of evidence investigating their overall efficacy and safety is warranted. However, given that the breadth and scope of the literature are vast and diverse before a higher level of synthesis is undertaken, the process of undertaking a systematic scoping review provides an opportunity to catalogue and summarise the current literature to date [29].

Therefore, this scoping review aimed to systematically map the available literature to identify the type and frequency of the use of nutraceutical and micronutrient supplementation for the management of PCOS features, including anthropometric, metabolic, reproductive and psychological. This review provides an overview of the breadth of the current available evidence and identifies existing gaps to direct future research, including the need for primary studies, systematic reviews and meta-analyses.

## 2. Materials and Methods

This scoping review followed a systematic approach, with the protocol registered at Open Science Framework registries (https://doi.org/10.17605/OSF.IO/UJXHP; accessed on 6 April 2022). The search strategy and methods for extracting data were congruent with the Preferred Reporting Items for Systematic reviews and Meta-Analyses extension for Scoping Reviews (PRISMA-ScR) checklist [30].

### 2.1. Search Strategy

The original search was conducted in April 2022 with no date restrictions applied. An additional search was repeated in October 2023 to capture the recently published literature. Databases including CINAHL, Cochrane reviews, Medline, PsycInfo, Scopus and LILACS were searched for relevant papers reporting on the use of nutraceuticals or micronutrients in women with PCOS. International trial registration databases were also examined for newly published papers. Finally, reference lists of all included papers were hand-searched for additional eligible papers. The MeSH search terms included ‘Stein Leventhal’, ‘Polycystic Ovary Syndrome’, ‘PCOS’, ‘PCOD’ and other variations. The search comprised various nutraceuticals and micronutrients, including an extensive list of individual vitamins, minerals, micronutrients and extracts of food, animals, or plants, in addition to overarching terms such as nutritional supplements, nutraceuticals, micronutrients, bioactive compounds, flavonoids, polyphenols, diet and mineral. Terms for PCOS and nutraceuticals/micronutrients were then combined to identify relevant papers. A complete search strategy for the Medline database is presented in Appendix A.

### 2.2. Eligibility and Exclusion Criteria

The eligibility criteria were determined by applying the PICOS (participants, intervention/exposure, comparator, outcome and study design) framework detailed in Table 1. Briefly, articles were included if they reported on the efficacy or association between nutraceutical and/or micronutrient supplementation on features of PCOS in women (≥18 years) with a diagnosis of PCOS reported by the study investigators. When no age was specified, articles were included if the study related to assisted reproductive treatments (ART) or the mean age (±standard deviation) was reported as ≥18 years. For the present review, we defined nutraceuticals as nutrient derivatives or extracts of plants, animals or foods that are used to elicit a potential pharmacological response, which includes vitamins, minerals, herbs and proteins [24]. Micronutrient supplements are defined as nutritional supplements that are used to improve nutritional status and/or mitigate dietary deficiencies (e.g., vitamin and mineral supplements). Features of PCOS included measurable outcomes classified according to four distinct characteristics: anthropometric, metabolic, reproductive and psychological. Studies were excluded if they were conducted with participants <18 years old or if study investigators did not report a PCOS diagnosis. Narrative reviews, case studies, editorials, conference abstracts and articles where the full text could not be accessed were also excluded. Due to funding limitations, papers not written in English were also excluded. Lastly, studies investigating the effects of individual food items or described as traditional medicines, such as Chinese or Ayurvedic medicines or decoctions, were excluded from the present review.

### 2.3. Study Selection and Screening

Results from the search strategy were uploaded into the systematic review data management software, Covidence (Veritas Health Innovation, Melbourne, Australia). Once duplicates were removed, a two-stage screening process was undertaken. Firstly, a duplicate screen of titles and abstracts was undertaken independently by seven authors (NS, AV, EM, LM, AM, CE, VR). In the second phase, full-text articles were screened independently by the same seven authors, with one author (NS) conducting a duplicate screening of all full-text articles. Any discrepancies that arose from the two screening processes were resolved by discussion and consensus, with consultation among all authors where needed.

### 2.4. Data Extraction and Synthesis of Results

Data extraction tables were developed by one author (NS) and adapted using feedback from three authors (AV, EM, LM). Data were extracted for eligible studies using Microsoft Excel (version 16.68) independently by one author (NS) and verified for accuracy and completeness by two authors (AV and EM). Data extracted included study design, sample characteristics and setting, study aims, intervention characteristics (including use/type of nutraceutical or micronutrient), reported adverse events and outcomes related to PCOS features (anthropometric, metabolic, reproductive and psychological). Findings were consolidated and presented descriptively in accordance with the individual nutraceutical or micronutrient. It was beyond the scope of the present review to synthesise the evidence on the efficacy of individual nutraceuticals and/or micronutrients on features of PCOS. However, individual study findings from the included studies are presented in Appendix A.

## 3. Results

### 3.1. Study Selection

The initial search retrieved a total of 16,649 records, of which 4300 duplicates and 11,267 abstracts were excluded, leaving 1082 articles for potential inclusion. Of these, 765 articles were removed after the full-text assessment, including 57 studies that are currently under investigation for concerns regarding data integrity, leaving a total of 317 articles that met the inclusion criteria. An additional 27 articles were identified during the most recent search, resulting in a total of 344 articles included in this review. The results of the article selection process are presented in the PRISMA flow chart in Figure 1. Each paper was counted as a separate study.

### 3.2. Study Characteristics

Comprehensive study details, including individual study outcomes, are presented in Appendix A. The 344 studies consisted of 192 randomised controlled trials, 19 non-randomised controlled trials, 30 single-arm trials, 10 observational studies and 93 systematic reviews. All studies were published between 2003 and 2023 and included a total of 24,607 participants with sample sizes ranging from six (an assessment of alpha lipoic acid and chromium) [31] to 3602 (an assessment of Myo-inositol) [32]. Trial durations varied from six weeks [33,34] to two years [35]; however, the exact length was ambiguous in some studies as it was described according to time points or cycle numbers of ovulation induction, ART or menstrual cycles. Trials originated from 34 different countries, representing Asia, Africa, North America, South America, Europe and Oceania.

### 3.3. Diagnostic Reporting of PCOS

The diagnostic criteria for identification of PCOS applied in the primary studies included Rotterdam (n = 188; 74.9%), National Institutes of Health (NIH) (n = 11; 4.4%), Androgen Excess Society (AES) (n = 6; 2.4%) or biological and/or clinical descriptions such as oligo/amenorrhea, hyperandrogenism, presence of hirsutism or polycystic ovaries by ultrasound (n = 26; 10.4%). Twenty (8.0%) failed to explicitly report on what diagnostic criteria were used to identify a diagnosis of PCOS.

In the reporting of systematic reviews, diagnostic criteria for PCOS were reported as part of the inclusion criteria in less than half of the included reviews (n = 37; 39.8%). Such criteria included Rotterdam (n = 35; 37.6%), NIH (n = 15; 16.1%), AES (n = 8; 8.6%) and biological and/or clinical descriptions such as biochemical assessments or polycystic ovaries on ultrasound (n = 3; 3.2%), with some reviews including more than one set of diagnostic criteria. One study from China included unspecified diagnostic criteria reported as the guidelines for the diagnosis and treatment of PCOS in China [36].

## 4. Study Outcomes

### Type and Frequency of Nutraceutical and Micronutrient Supplements Examined against Features of PCOS and Reporting of Adverse Effects

Forty-one different nutraceutical and micronutrient supplements were examined across the included studies. The most studied supplements included inositols (n = 86 studies), vitamin D (n = 53 studies) and N-acetylcysteine (n = 27 studies) (Figure 2). Across all studies, reproductive features were most commonly examined (n = 285; 82.8%), followed by metabolic (n = 229; 66.6%) and anthropometric (n = 197; 57.3%). Only eight studies (2.3%) reported on psychological outcomes. A total of 52 studies (15.1%) examined the potential efficacy of nutraceutical and micronutrient supplements on ART outcomes. Almost half (n = 161; 46.8%) of all studies included in the present review reported on the occurrence of adverse effects (Figure 3).

Inositols (Myo-inositol or D-chiro-inositol, including inositols in separate or combination formulas) were investigated across 86 separate studies. Among these studies, the most commonly reported clinical features of PCOS included reproductive (95.3%), followed by metabolic (64.0%) and anthropometric outcomes (49.0%). No studies investigated the use of inositols on psychological outcomes in women with PCOS. One-third of studies (33.7%) evaluated the potential efficacy of inositols on ART outcomes. Adverse effects were reported in over one-third (38.4%) of these studies.

Vitamin D was investigated across 53 separate studies. Reproductive outcomes were the most commonly reported feature of PCOS (77.4%), followed by metabolic (67.9%) and anthropometric (39.6%) features. Only a limited number of studies reported on the efficacy of vitamin D on psychological features (1.9%) or were assessed on ART outcomes (5.7%). Adverse effects were reported in 24.5% of these studies.

N-acetyl cysteine was investigated across 27 separate studies. Among these studies, the most examined characteristics of PCOS were reproductive features (92.6%), followed by metabolic (48.1%) and anthropometric (37.0%) features. No studies using N-acetyl cysteine in women with PCOS investigated its potential efficacy on psychological outcomes, whereas almost one-quarter of studies (22.2%) evaluated N-acetyl cysteine on ART outcomes. Adverse effects were reported in 66.7% of these studies.

Omega 3 fatty acids were investigated across 25 separate studies. Among these studies, metabolic outcomes were the most commonly reported feature of PCOS (80.0%), followed by reproductive (68.0%) and anthropometric (64.0%) features. No studies investigating omega-3 supplementation examined the potential benefits on psychological outcomes, whereas 4.0% were assessed on ART outcomes. Adverse effects were reported in 28.0% of these studies.

Biotics (prebiotics, probiotics and synbiotics) were investigated across 17 separate studies. Among these studies, metabolic and anthropometric parameters were the most commonly reported feature of PCOS (70.6%), followed by reproductive (52.9%) and psychological outcomes (5.9%). No studies investigating biotics were assessed on ART outcomes. Adverse effects were reported in 41.2% of these studies.

Cinnamon was investigated across 14 studies. Among these studies, metabolic outcomes were the most commonly reported feature of PCOS (85.7%), followed by anthropometric (64.3%) and reproductive outcomes (57.1%). No studies investigating cinnamon examined its potential efficacy on psychological outcomes or were assessed on ART outcomes. Adverse effects were reported in 57.1% of these studies.

A combination of nutraceuticals and micronutrients (multi-nutrient formulations) were investigated across 14 studies. Among these studies, reproductive outcomes were the most commonly reported feature of PCOS (92.9%), followed by metabolic (71.4%), anthropometric (64.3%) and psychological outcomes (21.4%). Multi-nutrient formulations were assessed on ART outcomes in 7.1% of studies included in the present review. Adverse effects were reported in 58.3% of these studies.

Co-enzyme Q10 (ten studies), carnitine (nine studies) and curcumin (eight studies) were investigated across 27 separate studies. Among these studies, reproductive outcomes were the most commonly reported feature of PCOS (21 studies; 77.8%), followed by metabolic (18 studies; 66.7%), anthropometric (16 studies; 59.3%) and psychological outcomes (2 studies; 7.4%). Carnitine and co-enzyme Q10 were investigated for their efficacy on all four features of PCOS and in outcomes related to ART. Adverse effects were reported in at least two separate studies for each supplement.

Chromium, resveratrol and berberine were investigated across seven studies each. Among these studies, reproductive outcomes were the most commonly reported feature of PCOS (21 studies; 100%), followed by metabolic (17 studies; 81.0%), anthropometric (15 studies; 71.4%) and psychological features (1 study; 4.8%). Resveratrol was examined for its potential efficacy on all four features of PCOS. Berberine and resveratrol were also assessed in outcomes related to ART. Adverse effects were reported in more than half of all studies for each of the aforementioned supplements.

Melatonin (six studies), selenium (six studies) and green tea (five studies) were investigated across 17 separate studies. Among these studies, reproductive outcomes were the most commonly reported feature of PCOS (14 studies; 82.4%), followed by metabolic (11 studies; 64.7%) and anthropometric (10 studies; 58.8%) features. All three supplements were examined for their potential efficacy on all three of the aforementioned features; however, none examined the potential benefits for psychological outcomes. Melatonin was assessed in outcomes related to ART. Adverse effects were reported in at least one study for each supplement.

Vitamin E, phytoestrogens, quercetin and fennel were investigated across four studies each. Among these studies, reproductive outcomes were the most commonly reported feature of PCOS (12 studies; 75%), followed by anthropometric (11 studies; 68.8%) and metabolic outcomes (9 studies; 56.3%). All four supplements were examined for their potential efficacy on all the aforementioned features. However, no studies examined their potential benefits on psychological outcomes or in outcomes related to ART. Adverse effects were reported in at least one study for all supplements.

Vitamin B (three studies), nigella sativa (two studies), magnesium (two studies), fenugreek seed extract (two studies) and thylakoid-rich spinach extract (two studies) were investigated across 11 separate studies. Among these studies, reproductive and anthropometric features were the most commonly reported (nine studies; 81.8%), followed by metabolic features (eight studies; 72.7%). All supplements were examined for their potential benefits on all three of the aforementioned PCOS features. None of these supplements were examined for their potential benefits on psychological outcomes. Vitamin B was the sole supplement examined for its efficacy on outcomes related to ART. Adverse effects were reported in at least one study for all aforementioned supplements, except magnesium, which was not reported.

L-arginine, sumac, maitake mushroom, agnugol, vitex agnus, liquorice, zinc, vitamin K2, celery, garlic, green coffee, green cardamon, oligopinn, psyllium, ellagic acid and astaxanthin were each investigated in single studies. Among these studies, reproductive outcomes were the most commonly reported PCOS feature (13 studies; 81.3%), followed by anthropometric (10 studies; 62.5%) and metabolic (8 studies; 50.0%). Sumac, vitamin K2, celery, green coffee and oligopinn were each examined for their efficacy on all three of the aforementioned PCOS features. None of these supplements were examined for their potential benefits on psychological outcomes. Astaxanthin was the sole supplement examined for its efficacy in outcomes related to ART. All studies reported on adverse events, with the exception of the studies on astaxanthin and green cardamon, which were not reported.

## 5. Discussion

To the best of our knowledge, this is the first scoping review to systematically map the available literature to identify the type and frequency of the use of nutraceutical and micronutrient supplementation for the management of anthropometric, metabolic, reproductive and psychological features of PCOS. Our scoping review synthesises a large body of evidence across an extensive range of nutraceuticals and micronutrients for the management of PCOS. We identified 344 papers reporting on 41 different nutraceuticals and micronutrients. Our systematic mapping of the literature shows that most studies report on reproductive, metabolic and anthropometric features of PCOS, with very few studies reporting on psychological outcomes.

Over half of the studies included in the present review reported on reproductive, metabolic and anthropometric PCOS features, with the majority of nutraceuticals and micronutrients investigated for their efficacy in regulating reproductive and metabolic dysfunction in women with PCOS. This is unsurprising given the well-recognised clinical features and subsequent risk factors associated with PCOS, including menstrual dysfunction, infertility, hyperandrogenism [2,37,38]; glucose and insulin dysfunction; T2DM [39]; and cardiovascular disease [36,40]. Furthermore, women with PCOS are at greater risk of overweight and obesity, central adiposity [41], and longitudinal weight gain, all of which exacerbate the aforementioned clinical features and risk factors for PCOS [39]. Similar to our findings, a scoping review by Kite et al. [42] reported that anthropometric, metabolic and reproductive outcomes were the most commonly reported when examining the extent of the available literature on resistance training interventions in women with PCOS, and there was a paucity of studies investigating psychological outcomes.

Importantly, women with PCOS have a greater prevalence of negative emotional symptoms compared to those without [15,43,44]. Women with PCOS are three times more likely to experience depressive symptoms and five times more likely to experience symptoms related to anxiety than women without PCOS [15]. Moreover, some investigators have reported that the prevalence of depression and anxiety are as high as 37% and 42%, compared with 14% and 9% in healthy controls, respectively [15,45]. Although the 2023 International PCOS Guidelines [1] provide evidence for mental health screening and diagnosis, there is also a need to consider novel therapeutic adjunct interventions with high efficacy and low side effects, such as nutraceutical and micronutrient supplementation. The potential role of nutraceuticals for the treatment of mental health disorders is steadily gathering a substantive evidence base. Recently, the World Federation of Societies of Biological Psychiatry (WFSBP) taskforce identified over two dozen nutraceuticals and phytonutrients that have been evaluated for their efficacy on mental health disorders in the wider population in randomised controlled trials [46]. However, the taskforce also identified heterogeneity in the methodologies used across these studies, including a failure to report on details regarding the blinding process and a failure to standardise active constituents. Moreover, results from our scoping review further highlight that the available evidence base for the use of nutraceuticals for attenuating negative emotional symptoms and mood disorders in women with PCOS is scant, thus limiting clinical confidence regarding their utility as an adjunct therapeutic strategy.

In the present review, we identified heterogeneity in the diagnostic criteria used for the identification of PCOS and treatment regimens across the examined studies, particularly in relation to supplement dosage, formulations and intervention durations (Appendix A). For example, in the present review, we identified that ~8% of studies failed to report on the diagnostic criteria used to identify PCOS, which would make direct comparisons between studies difficult. Moreover, some investigations across all studies also stated that treatment regimens and doses were decided arbitrarily or based on previous interventions, suggesting therapeutic doses remain largely unknown. Importantly, this heterogeneity makes it challenging for the development of future systematic reviews or meta-analyses with appropriate inclusion criteria in order to avoid unsuitable comparisons being made between studies. Further, we also identified a number of studies that included multi-nutrient formulations, in particular MYO, folic acid and alpha-lipoic acid, thus making it difficult for future research to identify the efficacy of each individual nutrient on the specific outcomes of interest, which may dilute, overestimate or confound the potential therapeutic benefits of a given nutrient. Almost 40% (n = 16) of nutraceutical and micronutrient supplements included in the present review were assessed in singular studies only, with their mechanism of action and proposed benefit on PCOS largely unknown. Although examining the efficacy of nutraceutical and micronutrient supplementation was beyond the scope of this review, the mapping of our results highlights the need for future research to identify the proposed mechanism of action and establish therapeutic doses and benefits across a wide range of clinical manifestations of PCOS, before nutraceutical and micronutrient supplementation can be integrated into practice. This includes the need for robust systematic reviews and meta-analyses with strict eligibility criteria and assessment of heterogeneity between primary studies, particularly on the effects of supplementation on various features of PCOS where there is a paucity of higher levels of evidence.

We also identified that inositols (reported in 86 papers) were the most frequently studied nutraceuticals in women with PCOS. Chemically, inositols are identified as hexahydroxycyclohexanes and include a family of nine isomers [47]. One of these, myoinositol (MYO), has been identified as the most common in all biological systems and is naturally found in plant-based foods, including fruits, legumes, wholegrains and nuts [48]. Nevertheless, two of the inositols, MYO and D-chiro-inositol (DCI), through different mechanisms, play important roles in the regulation of blood glucose. Specifically, MYO is involved in the cellular uptake of glucose and reduces the release of free fatty acids from adipose tissue, whereas DCI is involved in glycogen synthesis [49]. As such, the biological rationale underpinning the use of inositols as a potential therapeutic strategy for the clinical management of PCOS is derived from their insulin-mimetic properties to reduce postprandial blood glucose, with glucose metabolism shifted toward glycogen synthesis by DCI and glucose catabolism by MYO [50,51]. The 2023 International PCOS Guidelines [1] highlight that inositol (in either form) could be considered as an adjunct therapy in PCOS management. However, specific types, doses, or combinations of inositol could not be recommended for women with PCOS due to the lack of a consistent evidence base. Moreover, at present, the current guidelines do not consider emerging evidence for the use of any other form of CM for the management of PCOS [1].

Evidence suggests that over half the adult population in Western countries consume nutraceuticals for their potential therapeutic benefit, with habitual consumers being women and those with chronic disease [21,52,53,54,55,56,57]. In the present review, we uncovered 41 different supplements examined for their efficacy on clinical features of PCOS. The 2023 International PCOS Guidelines also highlight that women taking any form of CM are encouraged to advise their healthcare professional [1]. This recommendation is not only based on the proposed efficacy of the supplement but, perhaps more pertinently, its potential to cause serious adverse effects, particularly given that most decisions to use supplements are made by consumers themselves rather than health or medical professionals [58]. Although the intake of dietary supplements is generally considered safe, they are not without risk; this is because dietary supplements can be brought to market without the support of robust clinical trials, and as such, there is a paucity of studies evaluating the potential for serious adverse effects [58]. In the present review, we report a lack of consistency using a systematic approach for the recording of all adverse effects in response to nutraceutical or micronutrient supplementation, with over half of the included studies failing to report on adverse effect data. Given the increased use of nutraceutical and micronutrient supplements amongst women with PCOS [21,22,53,54], and taking into account the rapid growth of new research exploring the potential efficacy of these treatments, well-documented data describing the potential for adverse effects must be prioritised before these adjunct therapies can be effectively integrated into clinical guidelines and practice.

## 6. Strengths and Limitations

Strengths of this review include being conducted in accordance with the PRISMA-ScR checklist, with the protocol also registered on the Open Science Framework registries. We also systematically mapped the breadth of the available literature to identify the type and frequency of the use of nutraceutical and micronutrient supplementation for the management of PCOS across 344 papers and identified existing gaps to direct future research, including the potential for systematic reviews and meta-analyses. Limitations include the lack of a critical appraisal of the included studies, given that our aim was to conduct a broad and comprehensive scoping review to map the extent of the existing literature. Secondly, a clear and accurate method of categorising nutraceutical and micronutrient interventions was difficult, as study interventions frequently contained multiple supplements and/or multi-nutrient formulations. Thirdly, due to resource limitations, we excluded studies that were not published in English or included supplements described as traditional medicines, resulting in the possible exclusion of relevant studies. Fourthly, using the search term ‘nutraceutical’ and related keywords was indeed challenging due to the lack of a consensus or standardised definition for nutraceuticals [59,60], allowing room for interpretation by the research team. Of note, traditional herbal supplements or traditional medicines such as Chinese or Ayurvedic were beyond the scope of this review and were therefore excluded but warrant future investigation. Lastly, although it was beyond the scope of our review, we did not examine the efficacy of each of the nutraceuticals and micronutrients on clinical features of PCOS. Nevertheless, on the basis of our systematic mapping, we have identified several recommendations (below) and priorities to further advance research and practice.

## 7. Recommendations

This scoping review identified 344 studies that assessed the breadth of nutraceutical and micronutrient supplementation for the management of anthropometric, metabolic, reproductive and psychological features of PCOS. In order to translate the extent of the data presented in this scoping review, several priorities for future research have been identified, including:Future primary studies on the efficacy of nutraceutical and micronutrient supplementation on psychological outcomes in women with PCOS.Identification of therapeutic doses of nutraceuticals and micronutrients is needed to elicit clinically beneficial effects on PCOS features and the mechanisms underpinning these effects.Robust systematic reviews and meta-analyses of existing studies, particularly on the effects of supplementation on features of PCOS where there is a paucity of higher levels of evidence.Future primary studies should utilise and report on standardised PCOS diagnostic criteria, consistent with the most recent 2023 International Guideline for the assessment and management of PCOS.Future primary studies should consider further standardised criteria (e.g., diet, dosage and duration) with suitable inclusion criteria (age range, BMI groups and specific phenotypes) to allow for the comparison of reliable results.A more rigorous approach to monitoring and recording adverse effects is needed.Further primary studies should be adequately powered and representative of the population to improve external validity and facilitate translation into clinical practice.

## 8. Conclusions

This scoping review has systematically mapped the available literature on the type and frequency of use of nutraceutical and micronutrient supplementation for the management of anthropometric, metabolic, reproductive and psychological features of PCOS. We identified a wide range of nutraceuticals and micronutrient supplements investigated for reproductive and metabolic dysfunction in women with PCOS. However, few studies investigated psychological outcomes. We also reported a lack of consistency and poor reporting of the potential for adverse effects in response to nutraceutical and micronutrient supplementation. Several priorities for future research have been identified, which will provide a clearer understanding of the potential mechanisms and therapeutic doses needed to elicit a clinical benefit on all PCOS features. Until such research is positioned, it remains challenging to integrate nutraceutical and micronutrient supplementation as a potential adjunct therapy into clinical practice.

## Figures and Tables

**Figure 1 biomedicines-11-03349-f001:**
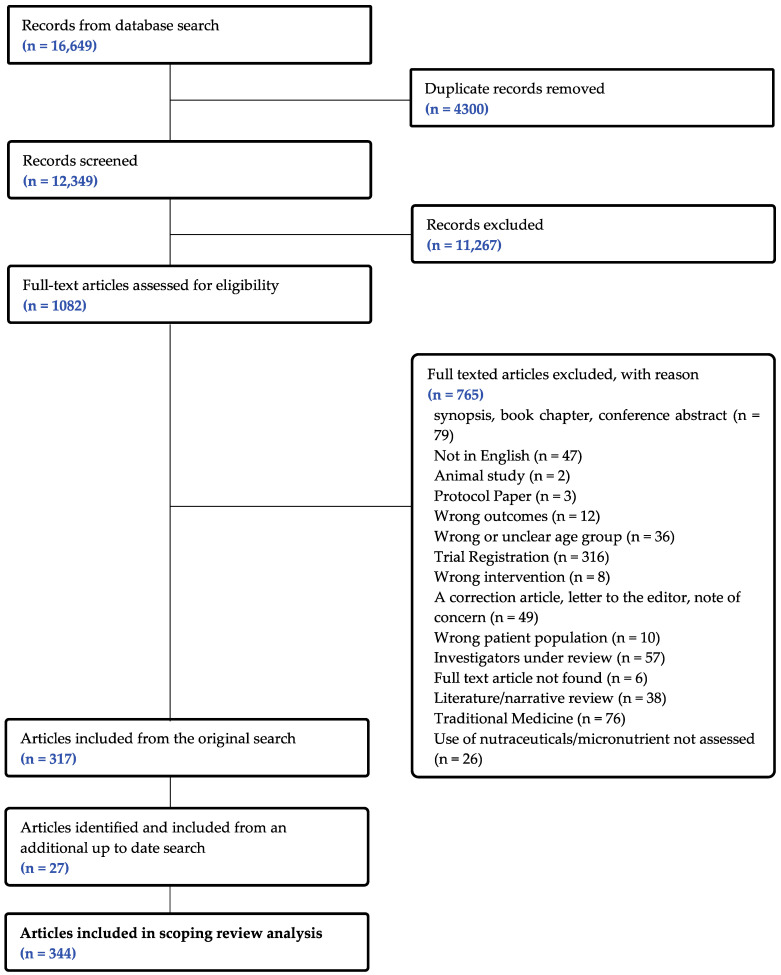
PRISMA flow diagram of study selection.

**Figure 2 biomedicines-11-03349-f002:**
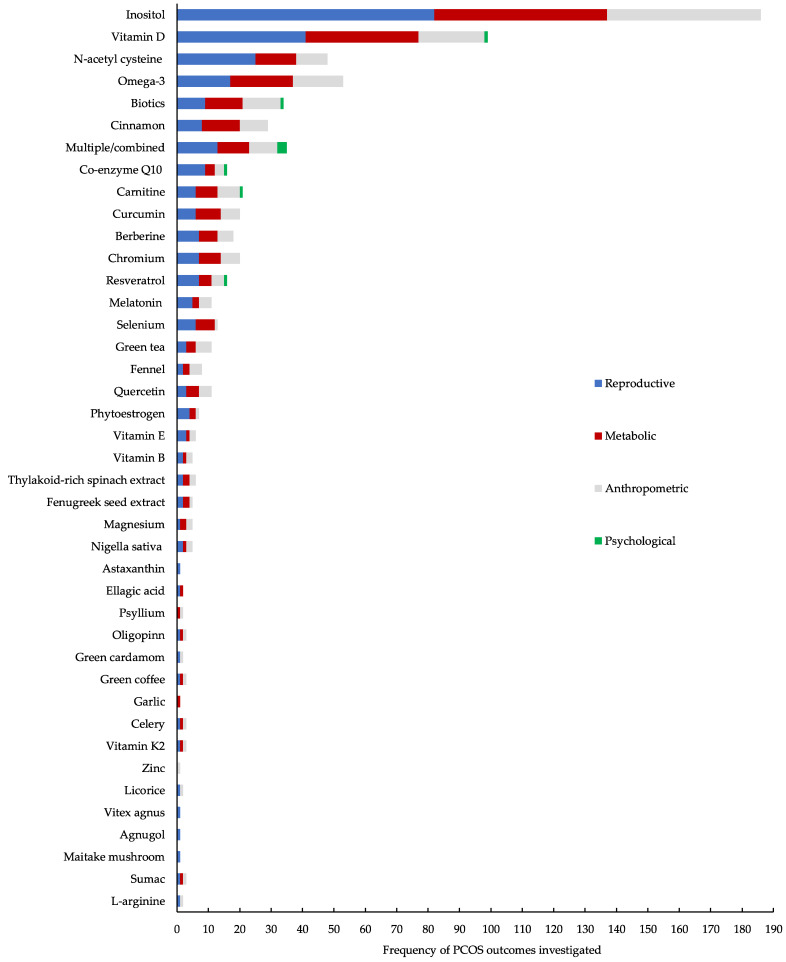
Frequency and type of nutraceutical and micronutrient supplements used for the management of reproductive, metabolic, anthropometric and psychological features of polycystic ovary syndrome.

**Figure 3 biomedicines-11-03349-f003:**
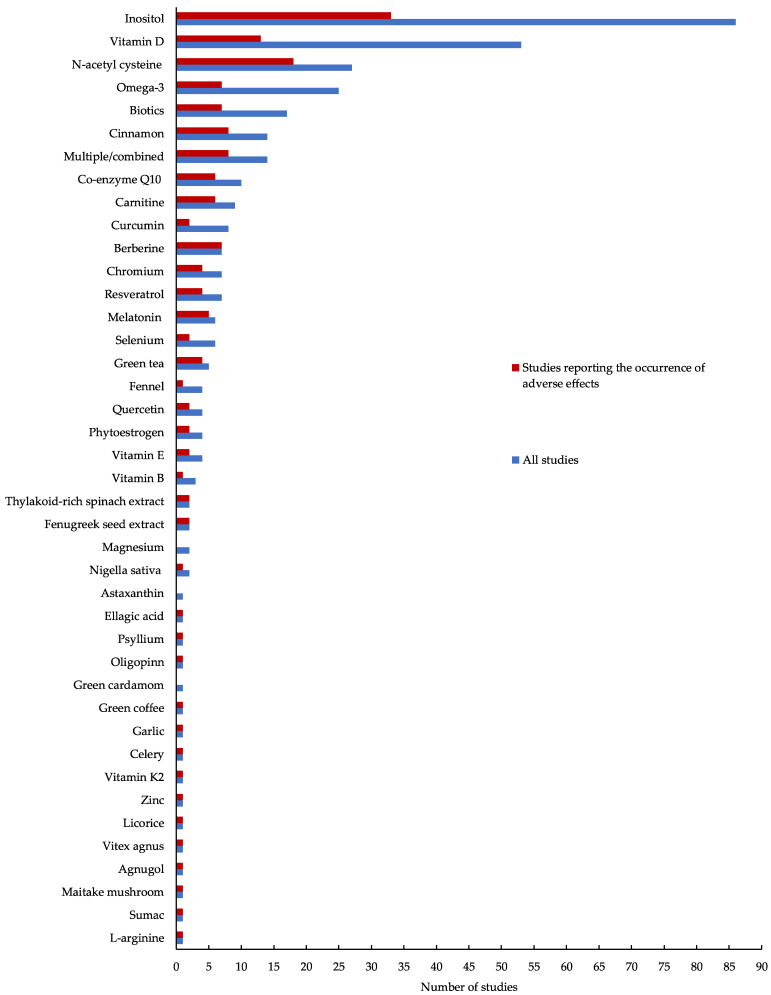
Frequency of reported adverse events.

**Table 1 biomedicines-11-03349-t001:** Inclusion and exclusion criteria presented against the PICOS framework.

	Inclusion Criteria	Exclusion Criteria
P	Women, ≥18 years, with a PCOS diagnosis reported by the study investigators	Women, <18 years, with no reported PCOS diagnosis
I	Nutraceutical or micronutrient supplement/s	Traditional medicines, decoctions or individual food items
C	With or without a comparator/control	
O	Assessment of PCOS features categorised by either anthropometric, metabolic, reproductive or psychological features	Outcomes assessed are outside of the categories of PCOS features, such as inflammation and blood pressure
S	Clinical trials, observational studies and systematic reviews	Case studies, narrative reviews, editorials or opinion pieces, conference abstracts, articles where full text was not available and languages other than English

## Data Availability

The datasets generated during and/or analysed during this scoping review are available from the corresponding author upon reasonable request.

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
