# Peer review of "Type and Frequency in Use of Nutraceutical and Micronutrient Supplementation for the Management of Polycystic Ovary Syndrome: A Systematic Scoping Review"

_biomedicines, 2023, doi:10.3390/biomedicines11123349_

Round 1
Reviewer 1 Report
Comments and Suggestions for Authors
In this review paper Scannell et al. maps the available literature to determine the type and frequency of use of dietary supplements and micronutrients to treat anthropometric, metabolic, reproductive and psychological characteristics of PCOS.
The topic of the review is interesting as there are gaps and inconsistencies in the literature regarding the treatment of PCOS.
In my opinion, the review does not provide a critical view of the authors, although it does make some recommendations for future research in the treatment of PCOS.
Thus, the Authors should discuss in a critical manner the main results emerging from the literature analysis, their strengths and limitations. In this version, the article is only a descriptive analysis of the studies reviewed.
Reviewer 2 Report
Comments and Suggestions for Authors
The authors had made great efforts to summarize the available studies on nutraceuticals in managing PCOS. Given the heterogeneity of the studies, the outcomes of this manuscript is unexpected. Have the authors attempted to focus only on the reproductive outcomes alone to derive more focused outcomes? This would enrich the manuscript substantially as the broad approaches on how the nutraceuticals can potentially manage PCOS is interesting and definitely worth in depth study.
Reviewer 3 Report
Comments and Suggestions for Authors
Comments: Manuscript ID: 2758273 , Title: Type and Frequency in Use of Nutraceutical and Micronutrient 2 Supplementation for the Management of Polycystic Ovary Syn- 3 drome: A Systematic Scoping Review
General comments
The manuscript is well-prepared and presents appropriate literature on the subject. Some aspects could be improved. After reading it, I have the impression that the review's authors focus very much on the issues "Study selection and screening", which is essential but obscures the conclusions from the obtained data. The manuscript has the structure of an original article, not a review article. In a non-exhaustive way, the discussion section presents the conclusions from 344 original works on PCOS. I am pleased with the presentation of the "limitations" and "recommendations" sections; they are necessary for this type of analysis of research results.
A list of my details and suggestions is below.
Specific comments
Materials and methods
Line 142-150: „2.3. Study Selection and Screening” this secetion is not revelant it is part of Author contribiution (Line 481..).
Line 142-150: „2.3 . Data extraction…”. Line 152-154 delete please. Line 155-166 : „Data extracted….” It shoud be limited.
Results:
3.1 Study selection. Line 177: I don't understand the reason for presenting Figure 1. It mostly presents information about rejected results
3.2. Study Characteristics: please delete lines 190-195
Lines 226-315 combine a lot of repeated information about the % distribution of the factor under study between each group. Perhaps the authors will consider presenting this in a table. It will be more readable.
Discussion:
The Authors focus mainly on presenting the results regarding the use of Inositol in PCOS patients. It may be worth expanding the discussion to include other substances such as vitamin D, N-acetyl cysteine or Omega-3 acids that are also common supplementing. . This section should be more related to the topic of the presented manuscript. It has a lot of general, overall information, that is important in helping to treat PCOS but is not so focused on the aim of this review.
The fragment, "In the presented..."; lines 398-414, will better suit after " therapeutic strategy"; line 358.
Reviewer 4 Report
Comments and Suggestions for Authors
This is an interesting scoping review with adequate novelty. However, several points need to be improved before reconsideration.
- The authors shold be report in the introduction section the literature gap that it is exist on this topic, reporting also the type of their review, e.g. that is a scoping review.
- The Methids and Materials section is very well-written and well-organized by the authors.
- I think that the Table 1 could be included into the text of the manuscript and not as supplementary material.
- From my pont of view
- In the introduction section, in the first paragraphs, the authors should a bit more recent bibliography from the last 2-3 years.
- The term "nutraceutical" remain a bit questionable so far. So, a short decription of this term should be described at the first mention of this term.
- At the last paragraph of the introduction section, the authors should further and briefly emphasized the literature gap that their scoping review will cover.
- From my point of view, I suggest to delete the term of nutraceuticals in both from the title and the text, since all the manuscript focused ton micronutrients supplemmments.
- Moreover, the term of nutraceutical is a bit confusing sicne there is not a sufficient international definition for them. Other terms such as micronutrients anf phytochemicals are more suitable and well-recognized.
- In general, the authors should include more references and especially more recent references.
Reviewer 5 Report
Comments and Suggestions for Authors
the systematic review submitted for review is carefully prepared and contains many references. you could consider dividing it into two parts
1. My first comment concerns the large number of references, including old ones. I suggest focusing on articles dating back 10-15 years, especially since some reviews and meta-analyses written by other authors were not included.
2. when writing the review, the authors took into account the PRISMA requirements http://www.prisma-statement.org/. The Flow chart is a sight to behold
3. The review did not cover several important aspects
A - the type of diet that should be the basis for supporting treatment, including Low GI and reduction, adding a paragraph is required
-Heliyon. 2021 Nov 9;7(11):e08338.doi:10.1016/j.heliyon.2021.e08338.
B- Vitamins Soluble in Water
-Nutrients. 2021 Feb 26;13(3):746. doi:10.3390/nu13030746.
C- licorice - Herbs supporting the treatment ran out
-BMC Compliment. Altern Med. 2014, 14, 511 DOI: 10.3390/nu13030746
-Front Endocrinol (Lausanne). 2019 Jul 18:10:484. doi: 10.3389/fendo.2019.00484
5. Written reviews of the topic were also omitted
-nutraceuticals and lifestyle review
Nutrients. 2021 Jul 18;13(7):2452. doi: 10.3390/nu13072452
-melatonin review DOI: 10.18502/ijrm.v17i12.5789
6. in the conclusions and abstract there is no specific summary regarding the beneficial supplementation in PCOS, no specific doses (range) are included
7. It is worth adding a drawing illustrating the benefits of supplementation on individual metabolic effects along with the doses, bearing in mind the phenotypic diversity in PCOS
best regards
Round 2
Reviewer 1 Report
Comments and Suggestions for Authors
If Biomedicine accepts "scoping reviews" the manuscript can be accepted for publication.
Reviewer 3 Report
Comments and Suggestions for Authors
I accept the Authors' comments and recommend the manuscript for publication.